

# Metagenomic analysis exploring taxonomic and functional diversity of soil microbial communities in Chilean vineyards and surrounding native forests

Luis E. Castañeda[1] and Olga Barbosa[1,2]

[1] Instituto de Ciencias Ambientales y Evolutivas, Facultad de Ciencias, Universidad Austral de Chile, Valdivia, Chile
[2] Instituto de Ecología y Biodiversidad (IEB-Chile), Santiago, Chile

Corresponding authors
Luis E. Castañeda,
lecastane@gmail.com
Olga Barbosa, olga.barbosa@uach.cl

## ABSTRACT

Mediterranean biomes are biodiversity hotspots, and vineyards are important components of the Mediterranean landscape. Over the last few decades, the amount of land occupied by vineyards has augmented rapidly, thereby increasing threats to Mediterranean ecosystems. Land use change and agricultural management have important effects on soil biodiversity, because they change the physical and chemical properties of soil. These changes may also have consequences on wine production considering that soil is a key component of *terroir*. Here, we describe the taxonomic diversity and metabolic functions of bacterial and fungal communities present in forest and vineyard soils in Chile. To accomplish this goal, we collected soil samples from organic vineyards in central Chile and employed a shotgun metagenomic approach to sequence the microbial DNA. Additionally, we studied the surrounding native forest to obtain a baseline of the soil conditions in the area prior to the establishment of the vineyard. Our metagenomic analyses revealed that both habitats shared most of the soil microbial species. The most abundant genera in the two habitats were the bacteria *Candidatus Solibacter* and *Bradyrhizobium* and the fungus *Gibberella*. Our results suggest that the soil microbial communities are similar in these forests and vineyards. Therefore, we hypothesize that native forests surrounding the vineyards may be acting as a microbial reservoir buffering the effects of the land conversion. Regarding the metabolic diversity, we found that genes pertaining to the metabolism of amino acids, fatty acids, and nucleotides as well as genes involved in secondary metabolism were enriched in forest soils. On the other hand, genes related to miscellaneous functions were more abundant in vineyard soils. These results suggest that the metabolic function of microbes found in these habitats differs, though differences are not related to taxonomy. Finally, we propose that the implementation of environmentally friendly practices by the wine industry may help to maintain the microbial diversity and ecosystem functions associated with natural habitats.

## INTRODUCTION

Being one of the main drivers of global change, land use change affects many important ecosystem properties and functions (*Vitousek et al., 1997*). Land use change (also referred to here as land conversion) has consequences at ecosystem scales because ecological functions can be lost during the conversion of native habitats (*Griffiths & Philippot, 2013*). Particularly in Mediterranean biomes, land conversion has occurred at very rapid rates over the last decades (*Cincotta, Wisnewski & Engelman, 2000*; *Lauber et al., 2008*; *Underwood et al., 2008*). This is especially important given that Mediterranean ecosystems are classified as biodiversity hotspots due to the high diversity of plant species and high endemism in these areas (*Cowling et al., 1996*; *Myers et al., 2000*). Therefore, conservation programs are necessary to preserve the biodiversity of these ecosystems.

The Mediterranean climate is suitable for viticulture; subsequently vineyards are becoming important components of Mediterranean landscapes due to the development of the wine industry in these regions (*Hannah et al., 2013*; *Viers et al., 2013*). Land occupied by vineyards increased by 70% between 1988 and 2010 in New World Mediterranean zones (Chile, California, Australia, and South Africa) (*Viers et al., 2013*). By replacing natural landscapes and by simplifying the structure and composition of ecological communities (*Viers et al., 2013*), the expansion of vineyards threatens Mediterranean ecosystems. In addition, agricultural management (e.g., tillage, pesticide, and fertilizer applications) directly affects soil biodiversity by altering the physical and chemical properties of soil (*Pampulha & Oliveira, 2006*; *Jangid et al., 2008*). For instance, organically managed vineyards have higher soil microbial biomass and nematode densities than conventional vineyards (*Coll et al., 2011*). On the other hand, *Bevivino et al. (2014)* report that undisturbed soils have more stable bacterial communities throughout the change of seasons than do vineyards. This suggests that natural habitats are more resilient to environmental or human perturbations. Furthermore, soil biodiversity is very significant to wine production, which relies on soil and climatic conditions as key components of *terroir* (*van Leeuwen et al., 2004*; *Gilbert, van der Lelie & Zarraonaindia, 2014*).

The soil horizon is one of the most diverse environments on Earth; currently it is estimated that thousands of different microbial species inhabit 1 g of soil (*Delmont et al., 2011*; *Xu et al., 2014*). There is abundant evidence confirming the important role played by soil microorganisms in several ecosystem services such as erosion control, soil formation, nutrient cycling, and plant health (*Tiedje et al., 1999*; *Nannipieri et al., 2003*; *Garbeva, van Veen & van Elsas, 2004*; *Gardi et al., 2009*). However, soil microbial communities are not static and can change across agricultural practices and environmental gradients (*Bevivino et al., 2014*; *García-Orenes et al., 2013*). For instance, the addition of organic matter to managed soils increases fungal abundance and causes the microbial community structure to resemble that of undisturbed forest soil (*García-Orenes et al., 2013*). In addition, *Corneo et al. (2013)* report that microbial communities change across altitudinal gradients, where soil physical (e.g., soil moisture, clay content) and chemical (e.g., aluminum, magnesium, molybdenum, and boron) properties explain most of the altitudinal variation in soil communities.

The recent development of high-throughput sequencing techniques has allowed a deeper understanding of the microbial diversity of vineyard soils in different wine-producing regions around the world (*Corneo et al., 2013*; *Fujita et al., 2010*; *Zarraonaindia et al., 2015*). Although Mediterranean Chile is one of the most important wine-producing regions and the area occupied by vineyards in Chile has rapidly expanded (*Viers et al., 2013*), there are very few studies exploring the microbial diversity of these vineyard soils (but see *Aballay, Maternsson & Persson, 2011*; *Castañeda et al., 2015*). Recently, *Castañeda et al. (2015)* have shown that the soil bacterial communities in native forests and vineyards are similar, whereas the fungal communities differ between the habitats. This study employed T-RFLPs, which are reliable technique but do not provide deep taxonomic resolution or information about the metabolic functioning of the microbial community. In the present study, our goal is to describe the taxonomic diversity as well as the metabolic functions of bacterial and fungal communities present in forest and vineyard soils in Chile. To accomplish this goal, we assessed the taxonomic and metabolic diversity of soil samples from three organic vineyards in central Chile; we employed a shotgun sequencing approach, paying particular attention to species associated with viticulture and wine making. The organic vineyards sampled are relatively young (<10 years old) and are surrounded by natural landscapes. The surrounding natural landscapes are dominated by native sclerophyllous forests and shrubs, thus these ecosystems likely represent the soil characteristics of the area before the establishment of the vineyard. The knowledge of the soil microbial communities of native habitats could provide valuable information for the conservation management of vulnerable ecosystems (*Heilmann-Clausen et al., 2014*) such as for the Chilean Mediterranean region (*Mittermeier et al., 2011*; *Hannah et al., 2013*; *Viers et al., 2013*). Knowledge of microbial community dynamics found in this biome is scarce and metagenomic studies could provide a starting point for the conservation of microbial diversity and for the preservation of ecosystem functions provided by natural habitats (*Gardi et al., 2009*).

## MATERIALS AND METHODS

### Sampling

Soil samples were collected from three different organic vineyards and from neighboring sclerophyllous forest patches in central Chile; Ocoa (32°52′S–71°7′W), Leyda (33°34′S–71°22′W), and Colchagua (34°36′S–71°7′W). Samples were collected in March (during the harvest season) of 2012. The owners of the vineyards and the surrounding native forest patches granted all necessary permits to access the sampling sites: Seña Vineyards in Ocoa (Chile), Cono Sur Vineyards in Leyda (Chile), and Emiliana Vineyards in Colchagua (Chile; Table 1). The vineyards contain woody-perennial monocultures of *Vitis vinifera*, whereas the forest patches mainly contain *Cryptocarya alba*, *Peumus boldus*, *Quillaja saponaria*, *Lithraea caustica*, and *Acacia caven*, among other tree and shrub species.

In each vineyard, a plot near the forest patch was randomly selected. In each vineyard plot, five vines each separated from the other by 3.5 m were randomly selected. One bulk soil sample was collected at a distance of 5 cm from each vine stem; the soil samples were taken from the first 15 cm of the soil horizon using soil cores. This depth was
**Table 1 Descriptive information of each sampling site.**

|  | Ocoa, Chile | Leyda, Chile | Colchagua, Chile |
|---|---|---|---|
| Latitude | 32°52′S | 33°34′S | 34°36′S |
| Longitude | 71°7′W | 71°22′W | 71°7′W |
| Altitude (m) | 307 | 216 | 268 |
| Mean temperature (°C) | 14.7 | 16.2 | 14.6 |
| Precipitation (mm) | 354 | 457 | 731 |
| pH forest soil[1] | 7.87 | 6.86 | 6.34 |
| pH vineyards soil[2] | 8.1 ± 0.1 | 7.8 ± 0.5 | 7.5 ± 0.4 |
| Forest soil content (sand, silt, and clay) (%) | 73–16–11 | 67–22–11 | 47–37–15 |
| Vineyard soil content (sand, silt and clay) (%) | 56–38–16 | 61–26–13 | 61–27–12 |
| Soil taxonomy | Alfisol | Alfisol | Alfisol |
| Vine variety | Cabernet Sauvignon | Sauvignon Blanc | Syrah |
| Planting year (±SD) | 2002 ± 3 | 2006 ± 1 | 2001 ± 4 |

**Notes:**
[1] pH in forests was determined from a single soil sample.
[2] pH in vineyards was determined in each plot and the mean (±SD) is shown.

chosen because the majority of microbial activity is thought to occur within the upper 15 cm (*O'Brien et al., 2005*). The same procedure was performed in the adjacent forest patch, where five native trees and corresponding soil samples were randomly selected and collected as previously described. All collected samples were stored in a sterile bag and placed in a cooler with ice packs. During the same day, the 30 soil samples were transported to the laboratory where they were individually homogenized, sieved, and stored at −80 °C until DNA extraction was performed.

## Metagenomic sequencing

For a total of 30 soil samples (three vineyards × two habitats × five soil samples), DNA was extracted using the Power Soil DNA isolation kit (MoBio Laboratories Inc., Carlsbad, CA, USA) following the manufacturer's instructions. The quality of the DNA extracted was determined by electrophoresis using a 0.8% agarose gel. Furthermore, the DNA was quantified using a nanospectrophotometer (NanoDrop Technologies Inc., Wilmington, DE, USA).

For sequencing, the DNA extractions from each habitat (five samples) were pooled into one sample. Thus, one pooled vineyard sample and one pooled forest sample were sequenced for each vineyard (six samples in total). The concentration of DNA was assessed by fluorescence using the Quant-iT PicoGreen dsDNA kit (Invitrogen, Carlsbad, CA, USA); fluorescence was measured on a DQ 300 fluorometer (Hoefer Scientific Instruments, San Francisco, CA, USA). Following this, each metagenomic library was prepared using the 454 GS Junior Titanium Rapid DNA library preparation kit according to the manufacturer's instructions. Emulsion PCR (emPCR) was performed according to the Amplification Method Manual using a Lib-L kit. All steps involved in massive DNA sequencing were performed in the AUSTRAL-omics Core-Facility (Facultad de Ciencias, Universidad Austral de Chile) in a 454 GS Junior Titanium Series (Roche, Branford, CT, USA) following the standard protocol from Roche.

## Data analysis

The raw sequences of each of the six metagenomes were uploaded to the MG-RAST server at http://metagenomics.anl.gov (*Meyer et al., 2008*). The number of uploaded sequences ranged from 141,694 to 195,138 sequences for the forest soil samples and from 189,372 to 208,095 for the vineyard soil samples. After quality control was performed using MG-RAST, the number of retained sequences for the forest soil samples ranged from 114,120 to 131,618 with an average length of 442.7 bp, whereas 108,385–138,101 sequences with an average length of 445.3 bp were retained for the vineyard soil samples (see Table S1 for more detailed information). Taxonomic assignments were performed using the SEED database, and metabolic assignments were performed using the Subsystems database. For both types of assignments, we employed a maximum *e*-value of $1e$–5, a minimum identity of 60%, and a maximum alignment length of 15 bp. The accession numbers for the metagenomes in the MG-RAST server (http://metagenomics. anl.gov/metagenomics.cgi?page=MetagenomeProject&project=8742) are: 4565458.3, 4565459.3, 4565460.3, 4565461.3, 4565462.3, and 4565463.3. Rarefaction curves for each of the samples reached appropriate taxonomic depth as can be seen in Fig. S1.

For taxonomic analysis, the OTU table was downloaded from the MG-RAST server and analyzed in QIIME v1.9.1 (*Caporaso et al., 2010*). OTUs that matched the following criteria were removed from the OTU table: (1) OTUs matched to mitochondria, chloroplast, plant, or animal sequences; (2) OTUs observed fewer than 10 times; and (3) OTUs observed in fewer than two samples. The resulting OTU table was analyzed employing the vegan (*Oksanen et al., 2013*) and phyloseq (*McMurdie & Holmes, 2013*) packages in R (*R Development Core Team, 2016*). To standardize the number of sequences between samples, they were rarefied to 289,800 sequences. Venn diagrams were made to visualize which OTUs were shared between forest and vineyard soils using Venny 2.1.0 (*Oliveros, 2015*). Species richness, Shannon diversity, and Pielou evenness indices were estimated for each of the samples, and these indices were compared between habitats using a Kruskal–Wallis test. Beta diversity was estimated using Bray–Curtis dissimilarity, employing the vegdist function of the vegan package in R. Then, a permutational multivariate analysis of variance (PERMANOVA) was used to compare the microbial community structure between forest and vineyard soils; this was performed with 999 permutations using the adonis function of the vegan package in R. Finally, a canonical correspondence analysis (CCA) conducted in vegan was used to visualize the community structure.

To analyze the metabolic profiles, the relative abundances of reads in forest and vineyard soils were compared via a White's non-parametric *t*-test (*White, Nagarajan & Pop, 2009*) using the software STAMP (*Parks & Beiko, 2010*). Comparisons of metabolic profiles between habitats were performed using a PERMANOVA analysis; this was done using the adonis function of vegan in R. Finally, CCA conducted in vegan was used to visualize the functional-based community structure and the relationship between soil samples and functional categories.

Finally, raw datasets and specific analyses are available at the Figshare server (https://dx. doi.org/10.6084/m9.figshare.2058060.v2).

## RESULTS

### Taxonomical analysis

Metagenomic analyses using the SEED database showed that bacteria, followed by Eukaryota and Archaea, dominated the forest as well as the vineyard soil samples. The other sequences corresponded to viruses and unassigned sequences (Table 2). Among bacteria, Proteobacteria was the most abundant phylum both in forest soil as well as in vineyard soil; which was followed by Actinobacteria, Acidobacteria, Bacteroidetes, Firmicutes, and Planctomycetes (Table 2). However, we did not find significant differences in the relative abundances of these phyla (Table 2). By taking a closer look at the taxonomy, we found 4,104 bacterial OTUs (97% nucleotide ID) corresponding to 1,326 species, of which 87.1% were shared between habitats (Fig. 1). The 10 most abundant species were *Candidatus Solibacter usitatus* (pooled mean = 2.5%, $P = 0.83$), *Bradyrhizobium japonicum* (pooled mean = 2.5%, $P = 0.51$), *Rhodopseudomonas palustris* (pooled mean = 2.1%, $P = 0.51$), *Conexibacter woesei* (pooled mean = 1.9%, $P = 0.83$), *Candidatus Koribacter versatilis* (pooled mean = 1.7%, $P = 0.83$), *Gemmatimonas aurantiaca* (pooled mean = 1.5%, $P = 0.28$), *Sorangium cellulosum* (pooled mean = 1.4%, $P = 0.83$), *Mycobacterium tuberculosis* (pooled mean = 1.4%, $P = 0.51$), *Rhodopirellula baltica* (pooled mean = 0.9%, $P = 0.83$), and *Myxococcus xanthus* (pooled mean = 0.9%, $P = 0.51$). Nevertheless, there were no significant differences in the relative abundances of these dominant species in forest and vineyard soils. Conversely, the relative abundances of 36 OTUs were significantly different between habitats ($P < 0.05$); but all of these OTUs were found in very low relative abundances in both habitats. We also explored the presence of lactic acid (Lactobacillaceae and Leuconostocaceae) and acetic bacteria (Acetobacteraceae) in the forest and vineyard soils; lactic acid bacteria is known to positively affect wine production while acetic bacteria negatively affects production. Typically, these bacteria are found in low relative abundances in soil samples, but we expected they might be found in the sampled vineyards, being derived from the grape skins that are often used as compost. We found the presence of lactic acid bacteria including *Lactobacillus* (vineyard = 0.04%, forest = 0.03%, and $P = 0.51$) and acetic bacteria such as *Gluconobacter* (vineyard = 0.038%, forest = 0.041%, and $P = 0.51$) and *Acetobacter* (vineyard = 0.13%, forest = 0.12%, and $P = 0.82$).

For the Eukaryota domain, we focused on fungal OTUs, which were mainly related to the phyla Ascomycota and Basidiomycota (Table 2). We found 95 fungal OTUs (97% nucleotide ID) corresponding to 47 Ascomycota and 8 Basidiomycota species. Among the most abundant fungal-related OTUs, we found the Ascomycota *Gibberella zeae* (vineyard = 0.040% and forest = 0.0042%, $P = 0.83$), *Aspergillus fumigatus* (vineyard = 0.03% and forest = 0.05%, P = 0.13), and *Neurospora crassa* (vineyard = 0.026% and forest = 0.029%, $P = 0.28$). Exploring the presence of fermenting yeasts in soil, we found some OTUs related to *Saccharomyces cerevisiae* (97% identity), which were significantly more abundant in forest (0.004%) than in vineyard (0.002%) soils ($P = 0.046$). Another important group found in both habitats was the domain Archaea represented by its five phyla: Crenarchaeota, Euryarchaeota, Korarchaeota,

**Table 2 Abundances of taxonomic groups in forest and vineyard soils.**

| Taxa | Forest | Vineyard | P-value |
|---|---|---|---|
| **Archaea** | **0.454 ± 0.050** | **0.486 ± 0.077** | **0.83** |
| Crenarchaeota | 0.050 ± 0.007 | 0.048 ± 0.003 | 0.51 |
| Euryarchaeota | 0.376 ± 0.040 | 0.395 ± 0.054 | 0.83 |
| Korarchaeota | 0.004 ± 0.002 | 0.004 ± 0.002 | 0.83 |
| Thaumarchaeota | 0.021 ± 0.015 | 0.036 ± 0.019 | 0.28 |
| Unclassified | 0.003 ± 0.001 | 0.004 ± 0.001 | 0.66 |
| **Bacteria** | **90.34 ± 0.561** | **90.43 ± 0.073** | **0.72** |
| Acidobacteria | 5.118 ± 0.748 | 5.080 ± 0.841 | 0.83 |
| Actinobacteria | 20.49 ± 2.532 | 20.17 ± 1.732 | 0.83 |
| Aquificae | 0.098 ± 0.005 | 0.105 ± 0.009 | 0.38 |
| Bacteroidetes | 3.049 ± 0.502 | 3.369 ± 0.576 | 0.51 |
| Chlamydiae | 0.051 ± 0.014 | 0.050 ± 0.013 | 0.83 |
| Chlorobi | 0.315 ± 0.030 | 0.321 ± 0.034 | 0.83 |
| Chloroflexi | 1.918 ± 0.304 | 2.017 ± 0.250 | 0.83 |
| Chrysiogenetes | 0.015 ± 0.005 | 0.016 ± 0.005 | 0.83 |
| Cyanobacteria | 1.774 ± 0.236 | 1.819 ± 0.190 | 0.51 |
| Deferribacteres | 0.048 ± 0.006 | 0.048 ± 0.005 | 0.83 |
| Deinococcus–Thermus | 0.574 ± 0.043 | 0.599 ± 0.049 | 0.51 |
| Dictyoglomi | 0.035 ± 0.010 | 0.046 ± 0.002 | 0.27 |
| Elusimicrobia | 0.012 ± 0.002 | 0.016 ± 0.004 | 0.27 |
| Fibrobacteres | 0.006 ± 0.001 | 0.010 ± 0.003 | 0.08 |
| Firmicutes | 2.945 ± 0.296 | 3.313 ± 0.295 | 0.13 |
| Fusobacteria | 0.028 ± 0.003 | 0.030 ± 0.011 | 0.51 |
| Gemmatimonadetes | 1.465 ± 0.208 | 1.542 ± 0.180 | 0.28 |
| Lentisphaerae | 0.030 ± 0.010 | 0.037 ± 0.003 | 0.28 |
| Nitrospirae | 0.200 ± 0.018 | 0.193 ± 0.037 | 0.51 |
| Planctomycetes | 3.001 ± 0.062 | 3.425 ± 0.617 | 0.13 |
| Ptobacteria | 0.019 ± 0.003 | 0.021 ± 0.007 | 0.83 |
| Proteobacteria | 46.12 ± 0.245 | 45.12 ± 1.481 | 0.28 |
| Spirochaetes | 0.235 ± 0.019 | 0.228 ± 0.015 | 0.83 |
| Synergistetes | 0.054 ± 0.003 | 0.062 ± 0.010 | 0.13 |
| Tenericutes | 0.001 ± 0.001 | 0.002 ± 0.001 | 0.27 |
| Thermotogae | 0.104 ± 0.010 | 0.134 ± 0.025 | 0.13 |
| Verrucomicrobia | 2.224 ± 0.656 | 2.210 ± 0.249 | 0.83 |
| Unclassified | 0.419 ± 0.040 | 0.439 ± 0.042 | 0.28 |
| **Eukaryota** | **0.582 ± 0.162** | **0.434 ± 0.139** | **0.51** |
| Ascomycota | 0.462 ± 0.102 | 0.339 ± 0.131 | 0.28 |
| Basidiomycota | 0.032 ± 0.013 | 0.026 ± 0.003 | 0.83 |
| Unclassified | 0.088 ± 0.056 | 0.069 ± 0.014 | 0.51 |
| **Viruses** | **0.002 ± 0.008** | **0.001 ± 0.009** | **0.49** |
| **Unassigned/unclassified** | **8.624 ± 0.370** | **8.623 ± 0.115** | **0.51** |

**Note:**
Values are shown as percentage of abundance in each habitat (mean ± standard deviation) P-values were derived from a Kruskal–Wallis test.

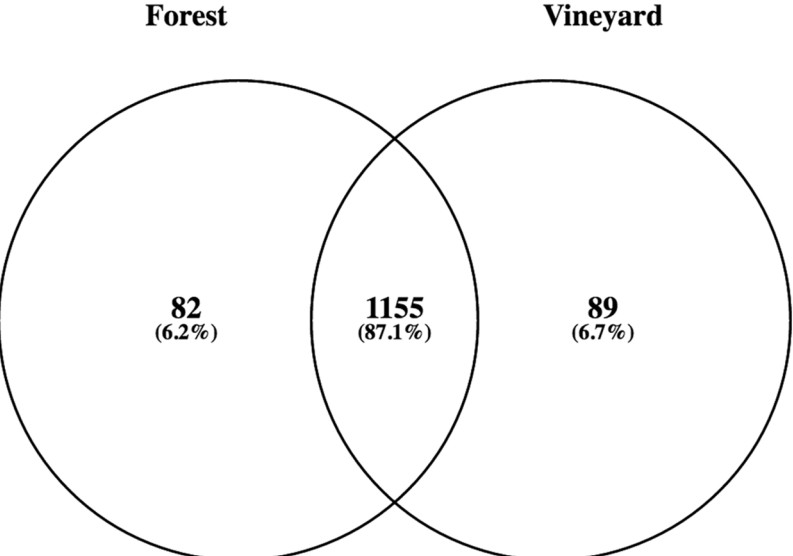

**Figure 1 Numbers and percentage of bacterial species found in forest and vineyard soils.** The number in the overlapping zone indicates how many species were shared between forest and vineyard soils, and the numbers in the non-overlapping zone indicate how many species were exclusively found in each habitat.

**Table 3 Diversity indices for microbial communities from forest and vineyard soils.** (A) Diversity indices for microbial communities and (B) Shannon richness related to functional categories likely associated with nutrient cycling.

|  | Forest | Vineyard | *P*-value |
|---|---|---|---|
| **(A) Index** |  |  |  |
| Richness | 5101 ± 14.7 | 5251 ± 28.6 | 0.05 |
| Shannon diversity | 7.175 ± 0.054 | 7.209 ± 0.075 | 0.51 |
| Shannon richness | 517.0 ± 15.4 | 530.6 ± 32.5 | 0.83 |
| Pielou evenness | 0.840 ± 0.007 | 0.843 ± 0.009 | 0.51 |
| **(B) Shannon richness** |  |  |  |
| Nitrogen metabolism | 287.8 ± 20.0 | 295.9 ± 11.2 | 0.51 |
| Phosphorous metabolism | 286.7 ± 11.9 | 258.5 ± 22.5 | 0.13 |
| Potassium metabolism | 183.6 ± 11.1 | 193.5 ± 12.7 | 0.28 |

**Note:**
Values are shown as percentage of abundance for each habitat (mean ± standard deviation) *P*-values were derived from a Kruskal–Wallis test.

and Thaumarchaeota. Of these, the phylum Euryarchaeota was the most abundant, but no significant differences were found between the Archaea of forest and vineyard soils (Table 2).

Microbial community analyses showed that vineyards had higher species richness than forests, while the Shannon diversity, Shannon richness, and evenness indices were not significantly different between the habitats (Table 3A). We also found that the microbial community structure did not differ between habitats (PERMANOVA, *P* = 0.45); this is illustrated in the CCA plot (Fig. 2).

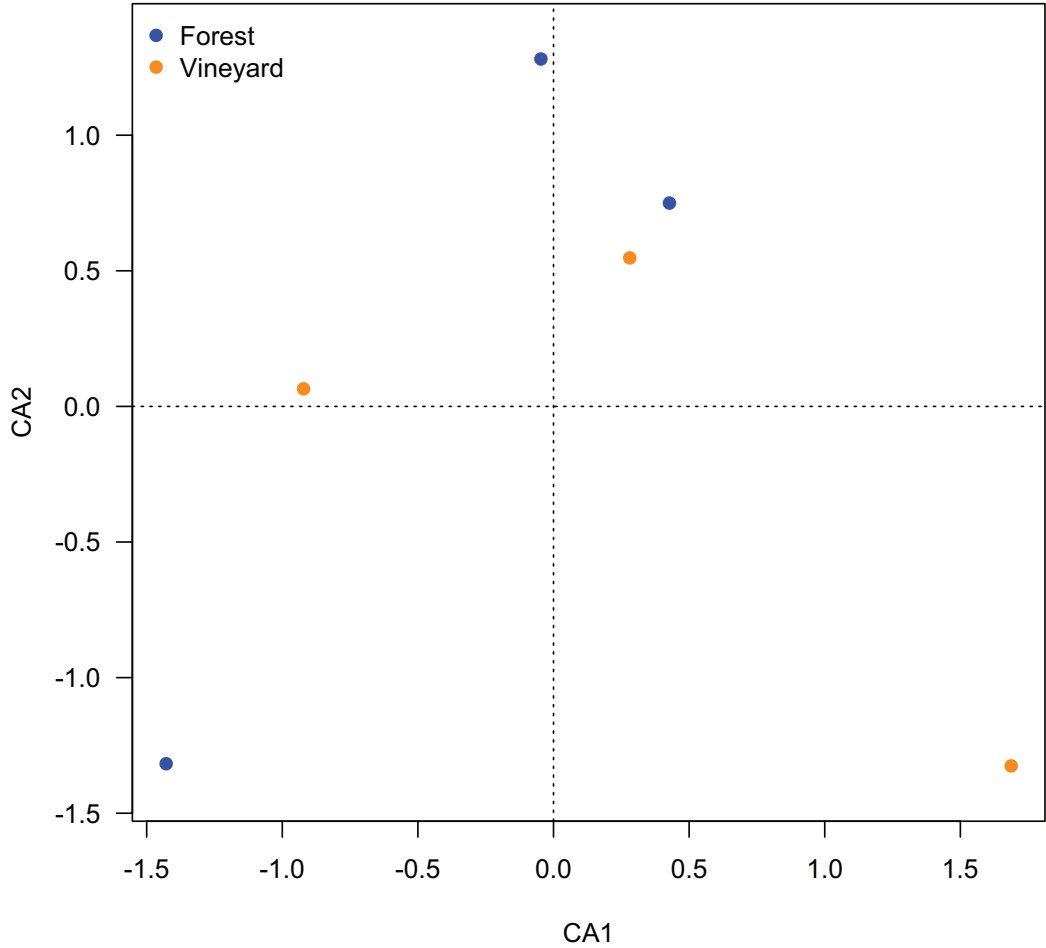

**Figure 2 Ordination plot for microbial composition in soils.** Ordination plot from the canonical correspondence analysis (CCA) based on the abundance of OTUs found the microbial communities found in forest and vineyard soils.

## Functional analysis

Functional metabolic categories related to microorganisms found in forest and vineyard soils are represented in Fig. 3. The most abundant functional categories included sequences related to carbohydrate metabolism (forest mean = 14.4% and vineyard mean = 14.6%), functionally coupled genes but with unknown function (i.e., clustering-based on subsystems) (forest mean = 14.0% and vineyard mean = 14.2%), and metabolism of amino acids and their derivatives (forest mean = 10.8% and vineyard mean = 10.6%). We did not find differences between habitats in the microbial communities' functional profiles at the SEED level-1 gene annotation (PERMANOVA, $P = 0.80$, Fig. 3), and any functional categories showed a clear association to any habitat type (Fig. 3). Conversely, we found that genes related to metabolism of amino acids and their derivatives ($P = 0.007$), fatty acid and lipid metabolism ($P = 0.024$), nucleoside and nucleotide metabolism ($P = 0.045$), and secondary metabolism ($P = 0.011$) showed significantly higher relative abundances in forest than vineyard soils (Fig. 4). On the other hand, only genes related to miscellaneous functions ($P = 0.033$) showed a significantly higher relative abundance in

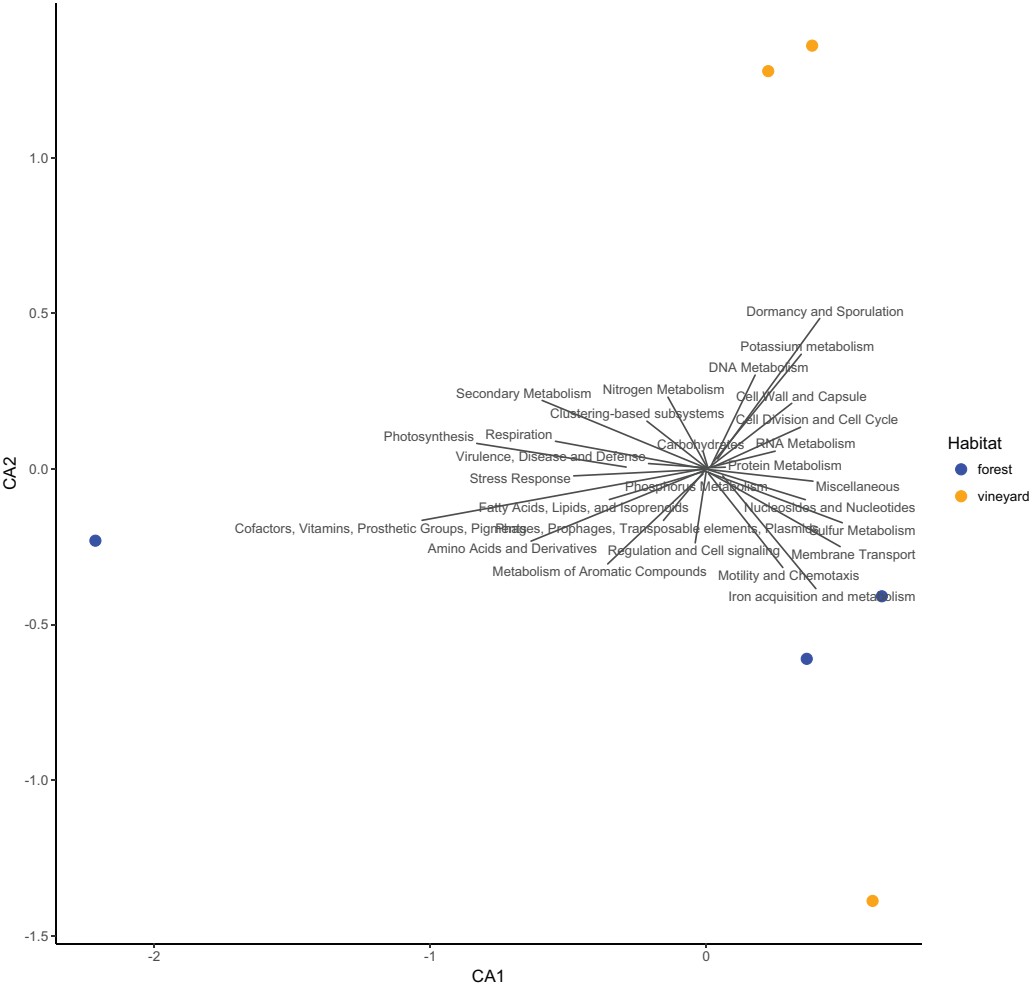

**Figure 3 Ordination plot for metabolic categories in soils.** Ordination plot from the canonical correspondence analysis (CCA) based on the relative abundances (points) and component loadings (text) of individual functional categories (SEED subsystem level 1) of the microbial communities found in forest and vineyard soils.

vineyard than forest soils (Fig. 4). Therefore, we explored the functional profiles of the categories that showed significant abundance between habitats, but we did not find significant differences in the microbial communities' metabolic profiles: amino acid metabolism (PERMANOVA, $P = 0.22$), fatty acid metabolism (PERMANOVA, $P = 0.23$), nucleoside and nucleotide metabolism (PERMANOVA, $P = 0.25$), secondary metabolism (PERMANOVA, $P = 0.40$), and miscellaneous functions (PERMANOVA, $P = 0.23$).

Additionally, we explored some functional categories that could be associated with nutrient cycling. From this, we found sequences related to sulfur metabolism (forest mean = 1.18% and vineyard mean = 1.15%), phosphorous metabolism (forest mean = 1.04% and vineyard mean = 1.05%), nitrogen metabolism (forest mean = 0.82% and vineyard mean = 0.80%), and potassium metabolism (forest mean = 0.30% and vineyard mean = 0.33%). The relative abundances of these functions were similar in forest and vineyard soils ($P > 0.1$). We also explored the SEED level-3 hierarchical gene

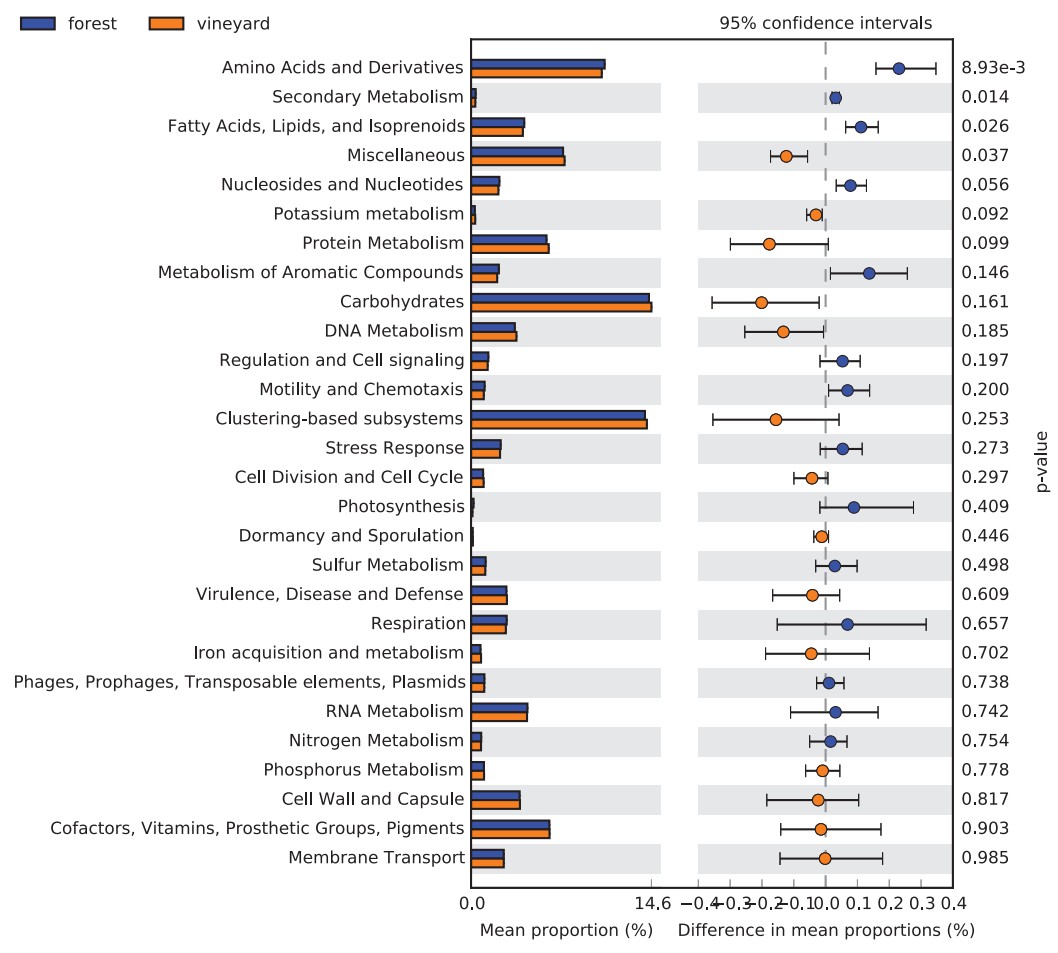

**Figure 4 Functional categories found in soil microbial communities.** Bar plot showing the mean proportion (%) of functional categories found in soil microbial communities based on the subsystem database. Points indicate the differences between forest and vineyard soils (blue and orange bars, respectively), and the values at the right show the *P*-values were derived from a White's non-parametric *t*-test (*White, Nagarajan & Pop, 2009*).

annotation. In general, assimilation of inorganic sulfur (overall mean = 0.37%), phosphate metabolism (overall mean = 0.54%), phosphorous uptake (overall mean = 0.20%), ammonia assimilation (overall mean = 0.38%), nitrate and nitrite assimilation (overall mean = 0.15%), and potassium homeostasis (overall mean = 0.28%) were the most abundant level-3 functions related to nutrient cycling. However, the relative abundances of these functions were not significantly different between forest and vineyard soils. We also estimated the species richness of OTUs related with some specific functional categories (level 1 of SEED subsystems hierarchy) such as nitrogen metabolism, phosphorous metabolism, potassium metabolism, and defense-related genes. However, we did not find any significant differences in alpha-diversity between habitats (Table 3B).

## DISCUSSION

Our analysis showed that bacterial-related OTUs had the highest relative abundance in both habitats. For soil environments, *Uroz et al. (2013)* have reported similar bacterial

abundances in organic and mineral soils: bacterial sequences accounted for ca. 94% of the total sequences. Proteobacteria are very common in soil environments and are related to a wide variety of functions involved in carbon, nitrogen, and sulfur cycling (*Spain, Krumholz & Elshahed, 2009*). The relative abundances of Proteobacteria found in the present study are similar to those previously reported for other soil types such as crops, forests, and grasslands (ca. 40% according to *Janssen (2006)*). Participating in carbon cycling and producing secondary metabolites, Actinobacteria are also dominant in soils (*Jenkins et al., 2010*). In our study, the most abundant bacterial genera in the soil samples were *C. Solibacter*, *Bradyrhizobium*, *Conexibacter*, and *Rhodopseudomonas*, which have been previously reported as dominant genera in several types of soil (*Delmont et al., 2011*; *Pearce et al., 2012*). Comparing bacterial phyla and genera, we did not find differences in relative abundances between forest and vineyard soils. Additionally, we found that diversity indices and microbial community structure were similar between forest and vineyard soils; this agrees with our previous work performed using T-RFLPs (*Castañeda et al., 2015*). Conversely, these findings differ from previous evidence suggesting that bacterial communities differ between forest and managed soils (*García-Orenes et al., 2013*). However, the relationship between microbial diversity and habitat disturbance is very complex and can depend on the degree of disturbance; some disturbed habitats can even exhibit higher diversity than forest systems (*Miura et al., 2016*; *Montecchia et al., 2015*).

Similar to the bacterial community results, fungal communities did not differ between habitats. Most of the fungi-related sequences were assigned to Ascomycota, outweighing other groups such as Basidiomycota, which only represented a small fraction of the total fungal sequences. At the species level, the most abundant fungal species was *Gibberella zeae/Fusarium graminearum*, a well-known plant pathogen that attacks cereals (*Bai & Shaner, 2004*). From a comparative point-of-view, we found similar fungal abundance between forest and vineyard soils. Interestingly, our previous work employing T-RFLPs showed that fungal community structure differed between forest and vineyard soils (*Castañeda et al., 2015*); this agrees with another T-RFLP-based study that shows that fungal diversity differs between native *Eucalyptus* forests and *Pinus* plantations in Australia (*Kasel, Bennett & Tibbits, 2008*). It should be noted that the lack of differences in fungal abundances in the present study might be related to the low representation of fungal sequences in the soil samples. Additionally, differences in taxonomic abundance could be limited to differences in functional taxonomic groups because taxonomic assignment was based on the SEED non-redundant protein database (for additional information see *Carrino-Kyker, Smemo & Burke, 2013*). Therefore, complementary approaches such as metatranscriptomic or amplicon-sequencing approaches should be employed to study soil eukaryotic communities to gain a deeper understanding of the ecology of these communities.

Microbes play important roles in several stages of wine production (*Mills et al., 2008*). For instance, fermenting yeasts are involved in alcoholic fermentation (i.e., the conversion of sugar into ethanol and carbon dioxide), and lactic acid bacteria perform malolactic fermentation (i.e., the conversion of malate into lactate) (*Fleet, 2003*; *Mills et al., 2008*).

Our data show the presence of lactic acid bacteria such as *Lactobacillus* and *Gluconobacter*, acetic bacteria such as *Acetobacter*, and the fermenting-yeast *S. cerevisiae* in the soil samples collected. Although these species are not usually common in soils, we decided to search for them because organic vineyards often use recycled grape skins (also known as pomace) as organic fertilizer. Knowing this, one would expect that some lactic acid bacteria and fermenting yeasts could colonize, or at least survive, in vineyard soils. Recently, *Zarraonaindia et al. (2015)* have reported that soil acts as a source of grape-associated bacteria, and thus with edaphic factors, soil can influence grapevine microbiota. However, the abundance of lactic acid, acetic, and fermenting microbes was relatively low compared to other dominant taxa. This suggests, contrary to what has been previously suggested (*Bester, 2005*; *Chen, Yanagida & Shinohara, 2005*; *Zarraonaindia et al., 2015*), that soil may not be a suitable ecological niche or reservoir for microorganisms important to wine production. It must be noted, however, that differences in methodological approaches may explain disparities between our findings and those previously reported: some studies have employed enrichment methods (*Bester, 2005*; *Chen, Yanagida & Shinohara, 2005*) or amplicon sequencing (*Zarraonaindia et al., 2015*), while shotgun sequencing (technique employed in the present study) could underestimate the abundance of fungal sequences. Future studies should evaluate the presence of enologically important microorganisms in surrounding native flora (i.e., leaves and fruits) to determine if these habitats are potential sources and/or reservoirs of microbial diversity relevant to wine production. This is particularly interesting due to the fact that high-quality wines are strongly associated with the concept of *terroir*, which encompasses regional characteristics such as climate and grape variety, and also gives special importance to soil and the interactions that occur with microorganisms (*Anonymous, 2010*). The fact that natural habitats can be potential reservoirs of microorganisms could safeguard the identity of *terroir* over time.

Most sequences obtained from forest and vineyard soils were related to the metabolism of carbohydrates and amino acids. This finding suggests that soil microbial communities are capable of degrading carbohydrates and playing an important role in the carbon cycle through organic matter and litter decomposition. Indeed, these results agree with the high relative abundance (ca. 12%) of genes related to carbohydrate metabolism previously reported for organic soils (*Uroz et al., 2013*; *Paula et al., 2014*). On the other hand, land-use changes alter the community structure of soil microorganisms and can have profound effects on ecosystem functions and processes (*Griffiths & Philippot, 2013*; *Paula et al., 2014*). In this sense, it has been reported that the land conversion of primary forests to long-term pastures has changed the microbial functional diversity of Amazon soils and especially so for genes related to carbon and nitrogen cycling (*Paula et al., 2014*). In the present study, we found differences in the abundance of genes related to the metabolism of amino acids and their derivatives, fatty acid and lipid metabolism, nucleoside and nucleotide metabolism, secondary metabolism, and miscellaneous functions. However, from a community perspective, we did not detect differences in the metabolic profiles nor did we find a difference in the number of microbial species related with each metabolic function. We also explored functional categories of genes related to nutrient metabolism

and potentially involved in nutrient cycling (*Fierer et al., 2012*). For instance, nitrogen-related genes represented 0.8% of the total functional reads, and the abundances of these genes did not differ between forest and vineyard soils. These abundance values are in concordance with previous studies, including environments enriched with nitrogen-fixing bacteria such as in soybean crops (*Mendes et al., 2014*). A plausible explanation for the lack of differences between habitats is that organic agriculture supplies nitrogen in its organic form (e.g., compost and manure) similarly to what occurs in forests; thus nitrogen could be available in the same chemical form in both habitats only in higher quantities in vineyards ($NH_4$ vineyard = 9.2 mg/kg and $NH_4$ forest = 4.2 mg/kg; $NO_3$ vineyard = 11.1 mg/kg and $NO_3$ forest = 7.2 mg/kg).

## CONCLUSION

We explored the taxonomic and functional diversity of microbial communities in Chilean vineyards using shotgun sequencing. We also analyzed the taxonomic and functional diversity of microbial communities in forest soils of the Chilean Mediterranean biome, one of the most threatened biodiversity hotspots in the world (*Myers et al., 2000*; *Viers et al., 2013*). Our metagenomic analyses revealed that the soil microbial communities of organic vineyards and native forests are similar, suggesting that taxonomic composition does not significantly differ between habitats. Conversely, some functional categories differed between forest and vineyard soils. These results could suggest that native forests surrounding vineyards act as microbial reservoirs buffering land conversion. However, additional research is needed to explore the role of landscape complexity and agriculture management on microbial communities in forest–vineyard agroecosystems. Finally, we propose that the implementation of environmentally friendly practices by the wine industry may help to maintain the microbial diversity and ecosystem functions related to natural habitats. This will not only preserve biodiversity but also help to maintain the typicity of wine, which is a valuable cultural and commercial characteristic.

## ACKNOWLEDGEMENTS

We thank Marlene Manzano and Fernando Alfaro for collecting soil samples and DNA extraction, Andrea Silva for advice during the metagenome sequencing, Juan Opazo for exploratory analysis on the sequencing data, and Juan Ugalde and Toshiko Miura for their advice on the metagenomic analysis. We also thank Elizabeth Cook and Emily Giles for their valuable suggestions and English editing of the manuscript draft. Finally, we thank Keith Crandall and the anonymous reviewers for their comments and suggestions that improved the quality of our manuscript.

### Funding

This work was funded by CONICYT PFB 23/2008 and ICM P05-002 through Instituto de Ecología and Biodiversidad (IEB-Chile). Luis E. Castañeda was partially supported by FONDECYT 1140066. There was no additional external funding received for this study.

The funders had no role in study design, data collection and analysis, decision to publish, or preparation of the manuscript.

**Grant Disclosures**
The following grant information was disclosed by the authors:
CONICYT PFB 23/2008 and ICM P05-002 through Instituto de Ecología and Biodiversidad (IEB-Chile).

**Competing Interests**
The authors declare that they have no competing interests.

**Author Contributions**
- Luis E. Castañeda analyzed the data, wrote the paper, prepared figures and/or tables, and reviewed drafts of the paper.
- Olga Barbosa conceived and designed the experiments, performed the experiments, contributed reagents/materials/analysis tools, and reviewed drafts of the paper.

**Field Study Permissions**
The following information was supplied relating to field study approvals (i.e., approving body and any reference numbers):

The owners of vineyards, whose property includes native forest patches, granted all necessary permits to access to the sampling sites: Seña Vineyards in Ocoa (Chile), Cono Sur Vineyards in Leyda (Chile), and Emiliana Vineyards in Colchagua (Chile).

**DNA Deposition**
The following information was supplied regarding the deposition of DNA sequences:

The accession numbers for the metagenomes in the MG-RAST server (http://metagenomics.anl.gov/mgmain.html?mgpage=project&project=mgp8742) were: 4565458.3, 4565459.3, 4565460.3, 4565461.3, 4565462.3, and 4565463.3.

**Data Availability**
Castañeda & Barbosa (2016): Metagenomic analysis exploring taxonomic and functional diversity of soil microbial communities in Chilean vineyards and surrounding native forests. Figshare.

https://doi.org/10.6084/m9.figshare.2058060.v2.

**Supplemental Information**
Supplemental information for this article can be found online at http://dx.doi.org/10.7717/peerj.3098#supplemental-information.

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
