# Peer review of "Metagenomic analysis exploring taxonomic and functional diversity of soil microbial communities in Chilean vineyards and surrounding native forests"

_PeerJ, doi:10.7717/peerj.3098_

## Round 0.1 · original submission · Major Revisions

As you know, we have now received a third review that is also fairly critical of your paper, but offers many good suggestions on improvement and is generally excited about the topic. The other two reviewers enjoyed your paper and thought it interesting and generally well done. However, both have some concerns and especially reviewer 2 would like to see a bit more done with the data in terms of exploring metabolic functions. Both reviewers noted a misalignment with the figures and tables and the references to them in the text. The third reviewer noted general issues with the text, grammar, etc. that need to be MUCH improved for your paper to move forward. Finally, you must submit your data to an online, open access repository and provide appropriate accession references for your data in the paper.

Reviewer 1 ·

Basic reporting

The manuscript by Castaneda and Barbosa describes the metagenomic analyses of the soils collected from the vineyard and forest sites. The authors have used shotgun metagenomic sequencing technique to study the microbial communities and their functional capacities. The manuscript describes the differences and similarities in two samples. I have some queries which needs to be clarified before the final publication.

Experimental design

Authors should provide the time (season) of the samples collected because time of sampling has substantial effect on the microbial communities. Also this could help the authors to explain, why eukaryotic diversity has been observed less in the shotgun metagenomic samples.

Validity of the findings

The authors have used standard procedure to study the microbial communities and their functional potential using MG-RAST. The data submitted to the MG-RAST server for the analysis. However, the data is not publicly available for the access and review process and author's should provide the data publicly available before the final publication.

Additional comments

1. Line number 174. The citation for the microbial abundances table is wrong. I think this is table 2.
2. Line number 212. The Figure 1 and Figure 2 is not included anywhere in the manuscript instead Table 3 and Table 4 is present which should be changed as Figures. Functional categories are shown in the Figure 2 and this citation should be corrected.
3. Line number 262 to 264. The sentence is not clear about "which molecular technique".

·

Basic reporting

The manuscript is logically structured and well written. The authors have adhered to recommended structures and provided sufficient background literature in the introduction. Appropriate citations for discussion have also been used. Accession numbers for metagenomic sequence data have been provided.

Experimental design

The authors have applied random sampling to obtain material for the study. For each vineyard or forest area, soil around 5 randomly selected vines or trees were sampled. These are relatively few sampling points and also few sampling sites to represent "Chilean vineyards". Therefore, I think the authors should rethink "Chilean vineyards" in the title. Also, 5 vines per vineyard? Maybe a diagrammatic representation to illustrate the coverage of the sampling strategy relative to the size of the vineyard or forest should be provided.

Validity of the findings

The data presented is statistically sound. However, there is misalignment between the tables and figures in text and the actual tables and figures presented. Overall, the data presented is very minimal and should perhaps be mined further to identify unique metabolic process if any between the two sites (i.e. at a genetic and enzyme level) especially for metabolic processes where there are significant differences.

Additional comments

Line 35- change "unlikeness" to unlikelihood
Line 40- replace "than" with "as"
Line 40 - 41, rephrase sentence starting with "additionally"
Line 135- replace "polled" with "pooled"
Line 136- insert (. and space) after vineyard area
Line 138-replace (.) with a comma before (Then) and make a continuous sentence rather than starting a sentence with "Then"
Line 330- replace "than" with "as"
Lactic bacteria should be changed to lactic acid bacteria throughout the document
In the results section the numbering of tables and figures is wrong (e.g. the data presented in the first paragraph is depicted in Table 2 not Table 1
There is reference to Figure 1 and 2 under functional analysis but the data is depicted under what is labelled as Table 4.
Also, what is presented as Table 3 and 4 should be Figures?

·

Basic reporting

See comments below (same comments are in the attached PDF)

Experimental design

See comments below (same comments are in the attached PDF)

Validity of the findings

See comments below (same comments are in the attached PDF)

Additional comments

I enjoyed reading over your manuscript – “Metagenomic analysis exploring taxonomic and functional diversity of soil microbial communities in Chilean vineyards and surrounding native forests” submitted to Peer J.

I think this manuscript is focused on an interesting topic which has implications for a better understanding of the microbial influence of native plants on those with agricultural importance, especially in the context of the model taxon of grape in the productive region of Chile.

Overall, I think this paper will be an important contribution to the literature, but as I see it now it has some severe issues and at the very least I would recommend that this paper face major revisions. I hope my comments here can help you craft a stronger contribution and provide more impact to the scientific literature. In its current state, this paper would not be a valuable contribution to the literature and I don't believe is in a publishable state.

==================================
Merits

The topic of this paper is both interesting and important in an ecological and enological sense. It has implications for the understanding of terrior as an influence on wine production and quality. This paper can contribute to this research area, but it needs some work to do so. As a result, I think this study has merits and would benefit both society, particularly in the context of those facing a changing climate and the preservation and protection of native lands while allowing for agriculture to be productive. This paper, after major revion, could advance the scientific knowledge in this area and provide the community with valuable information.

Critique

Unfortunately, I found your manuscript to be marred by many serious issues. I’ll outline the major issues here and then give a line by line commentary on some smaller concerns of mine.

General issues:

1. Poor writing and communication mar this paper. These problems include improper use of terms, poor grammatical structure, and misspelled words. I have gone over a few of these in the line-by-line comments below. Please have multiple people read the paper for content before you submit here or anywhere – it needs work. Numerous places are lacking commas and are not well punctuated. The are some long sentences that need to be simplified so that they make sense. There are a few “run-on” sentences also. I'm infuriated as my time is valuable and you should respect the reviewers and readers by actually putting some time into making sure your paper is well communicated. If the communication is so poor in the paper, how can we trust the science?

2. I don't understand why you don't record or characterize the plants adjacent to the vineyards. The plant taxonomy of the soils would surely inform the soil diversity. I just don't understand this lack of information with regards to the study.

3. Overall, I have some strong issues with how you have interpreted your findings and analyzed your data. First, the only analysis I am able to see that you did was upload your data to the MG-RAST server (but you do not show any of your output from that free analysis platform) and then run your data through the STAMP server (for output of Tables 3 and 4). There are minimal figures here (three data tables in the main paper), when metagenomic sequencing and analysis is a data intensive analysis. Nothing was done. I expect to see a short read component (which you just barely scratch the surface with) and a metagenome assembly and characterization component which you do not attempt. These are standard beginnings to analyses. You don't fully complete either analysis here. This lack of analysis makes the conclusions you do make skeptical and makes you sound like you do not understand what you are trying to do. For example:

Line 175-177: "Exploring the SEED database, we found 333 genera of which Prosthecochloris and Flouribacter were only found in forest soils, whereas Erwinia and Neorickettsia were only found in vineyard soils."

I don't see where you did standard metagenomic data analysis practices here, where you elimnated singletons and low abundance errors (you don't talk about how you clean or prepare your data) and this translates to two examples of you pointing out two genera (Prosthecochloris and Flouribacter) which are of such ridiculous levels in the reads that you accent their exclusivity.

Lines 309-311: "Higher abundances of genes related to ecological function such as metabolism of secondary metabolism and potassium metabolism were found in forest and vineyard soils, respectively"

Again, as above, what is your definition of "genes related to ecological function"? You are just taking the results that are spit out from STAMP and re-communicating them, but there is no analysis or interpretation in this paper. I have severe doubts with the findings you show.

4. There are no deposition numbers for the MG-RAST data that you provide.

Line By Line Comments:

Line 21: “historically related” – I do not understand what this means - that Mediterranean climates have traditionally been associated with wine production? This is awkwardly stated, also you should have a citation with many of these statements.

Line 22-23: "increased considerably threatening these Mediterranean ecosystems" - This also needs a citation, I am not aware of this as a strong argument agaist threatened biodiversity – I would expect it, but a citation would be good.

Line 24: Throughout the paper I would adhere to using the Oxford comma, as there are some confusing, as the grammar of this sentence: "Land use change and agricultural management affect soil biodiversity, changing physical and chemical properties of soil".

Line 26 - I do not consider terrior to be equal to "wine identity". Terrior is defined by Merriam Webster as "the set of all environmental factors that affect a crop's qualities, when the crop is grown in a specific habitat." I do not see how this connects to "wine identity". Perhaps a rewrite of this sentence is needed.

Line 26-27: "Here, we characterize the taxonomic and functional diversity of bacterial and fungal communities present in soil from vineyards in Central Chile" - I would not say that you do this. There is very little characterization in this manuscript. You either need to remove this sentence or actually characterize the taxonomic and functional diversity.

Line 30-31: "Our metagenomic analyses revealed that both habitata shared most of the soil microbial species" - this is not surprising at any level, especially on the basis of the level of characterization.

Line 31-33: "In general, bacteria were more abundant than fungi in both types of habitats, including soil-living genera such as Candidatus Solibacter, Bradyrhizobium and Gibberella" - yes, this has already been characterized in hundreds of studies, so this is not surprising. Why select for Solibater and Bradyrhizobium -- these were not the most common? Gibberella is a fungus.

Line 33-36: "Interestingly, we found presence of lactic bacteria and fermenting yeasts in soil, which are key during wine production. However, their abundances were extremely low, suggesting unlikeness of soil as a potential reservoir in Chilean vineyards." - I'm surprised by this - you found yeasts in abundances that we would expect, but why would you expect them to be in greater abundances when we already have literature that suggests these are late stage alcohol producing community members.

Line 37-39: "Regarding functional diversity, we found that genes for metabolism of amino acids, fatty acids, nucleotides and secondary metabolism were enriched in forest soils, whereas genes for metabolism of potassium, proteins and miscellaneous functions were more abundant in vineyard soils." - Poor grammar. Again, the use of commas would make these sentence clearer.

Lines 50-57: For these lines and throughout the paper, you are not consistent with how you are citing works. Sometimes with "NAME, et al., YEAR" or "NAME, et al. YEAR". Make sure every name is separated by a comma -- you don't do this. This, along with a lot of other things in the paper, come off as really sloppy and don't provide the appearance that you spent very long on both the study or the process of communicating it. Also, review the recommended formatting for Peer J.

Lines 59-60: "Mediterranean climate is suitable for viticulture, which historically has thrived in these areas" - Poor grammar. Also, what does "historically has thrived" means -- this should be re-written to be stated more clearly.

Lines 60-61: "During the last decades, land occupied by vineyards

61 has increased by 70% between 1988 and 2010" - too much redundancy here.

Line 63: should be "changing the physical" - you have done this a lot in the paper and leave out some key words of particular sentences. I would have someone read over this paper to fix for mistakes.

Line 73: "current information estimates the presence of 2,000 to 18,000 microbial genomes in one gram of soil" - You are either confused here or need to re-read the two citations you use here. This is not what either of these papers state.

Line 77: "soil microbial communities change across agricultural practices and environmental gradients" - do you mean the microbes change the practices and gradients or are changed by these? Again, I understand what you are trying to say here, but the writing is not clear and poor.

Line 79: "addition of organic matter increases the fungal abundance in managed soils" - This is
one study, but there are other oppositional studies with which to cite.

Line 84: "Recent development" - high-throughput sequencing is now more than a decade old.
This is no longer a recent development.

Line 91: "T-RFLPs is" - should be "T-RFLPs are" if you're just talking about the technique then
you have to re-write the sentence to not use the plural.

Line 91-92: "does not provide a deep taxonomic resolution" - neither does amplicon sequencing
or metagenomic sequencing in this context. There is no real difference. I would focus on the
functional aspect of your sentence if this is the angle you want to take with your discussion.

Line 97: Should be "are surrounded with natural landscapes"

Line 105-106: "the biodiversity hotspot, where the hotspot status" - awkward wording.

Line 135: "polled" should read as "pooled"

Line 136: "areaThe" should read as "area. The"

Line 168: "(mean = 95.97 % and 95.97 %, respectively)" - this can't be correct, the same amount
does not make sense in this context and reeks of typo here.

Line 177: Should read as "lactic-acid bacteria"

Line 195: Should read as "Among the domain Eukaryota"

Line 196: Should read as "the classes of Ascomycota and Basidiomycota"

Line 268: Should read as "approximately 0.5% of sequences"

Line 276: I think you are confused with the term "coincide"? Do you mean "agree"? You did not
study their habitat.

Line 284: Check the names of the people you cite throughout this paper -- you have mis-spelled
the authors name in the citation and throughout your manuscript.

Line 293: Should read as "important microorganisms for wine production"

Line 359: Should be "screening" and not "screenign"

Line 395: Should be "Japanese" and not "Japanise"

Lines 413-414, 421-422, 447-448, etc.: You have no paper title for these citations. Please check all the citations. You only have authors and journal here.

Final Note

I lastly just wanted to stress that I find this manuscript a very fascinating area of research and my comments are meant to constructively improve the content of the findings you present. Please do not take the comments personally, I just want to make sure your presentation and argument is a strong as it can be. I am sorry if my frustration came through in the review – I just don’t think this paper was ready to be submitted and it is a little frustrating to read a paper that needs so much work. It was a pleasure to review this paper, because I think the research area is extremely important, and, I hope my comments contribute to an overhaul of the paper with regard to clarity and improve the science. As with other papers in this area, I think this manuscript could represent a key contribution to the literature.

Warmest Regards,
Joshua R. Herr, Ph.D.
Assistant Professor, Department of Plant Pathology & Center for Plant Science Innovation, University of Nebraska, Lincoln, 68583-0722, USA

---

## Round 0.2 · Major Revisions

There appear to still be a large number of grammatical and English issues with your manuscript and both reviewers have commented on this. Please carefully check the next round.

Additionally, reviewer 3 makes a number of comments that require additional work, including replacing figure 4.

·

Basic reporting

No comments

Experimental design

no comments

Validity of the findings

no comments

Additional comments

The quality of the manuscript is vastly improved and it reads better.

·

Basic reporting

There are considerable changes to the manuscript which reflect an effort from the authors to clarify their writing. I find these changes to be well done. I apologize for my comments and my apparent frustration in reading the first draft, but I do not think the first draft was ready for submission. I find this subsequent draft to be in much better shape and the changes to the manuscript are noted and appreciated.

Experimental design

While I do not find the experimental design to be overly robust, the design is adequate for the study. I would have liked to see the authors record the differences in host leaf in the surrounding forests, but that can be conducted in a future study.

Validity of the findings

I welcome the changes to the manuscript in the additional three measures of diversity. I do think there are more analyses that you are lacking in a standard metagenomics study than you show, and I find this frustrating, but you have made an improvement on your first draft. I would include an assessment of taxonomic and functional diversity through metagenomic sequence read assembly, an assessment of 16S and other markers via the assembly, and an assessment of functional diversity through annotation of the assembled reads. I understand that these are time consuming analyses, but these are now, as of the last 5 years, standard analysis for metagenomic sequencing.

The data is available on MG-RAST, thank you for updating that -- I think this is vital for your study and should include the raw data. I am wondering why you did not include the automated analyses included from MG-RAST? This is already conducted (I found it online) and it's inclusion would improve the graphics of your manuscript.

Additionally, I find figure 4 to be unacceptable. I understand that you are using a standard program for the analysis, but with a little tweaking of the code inside the tool, you can make sure none of the text on your figure is overlapping. This can be done in one line of code with little time. I do not think that this figure in it's present form is in publishable shape.

Additional comments

The above comments will supplement my previous comments on the first draft. Thank you for your time in revising the first draft.

---

## Round 0.3 · accepted · Accept

Thanks again for your careful revisions. I think we are good to go now.

Reviewer 1 ·

Basic reporting

No comment

Experimental design

no comment

Validity of the findings

no comment

Additional comments

The quality of manuscript has been improved significantly, however, still the manuscript has some minor typographical errors and needs to be checked carefully.
such as I found one at line number 226: “lactic acid is known………“ should be “lactic acid bacteria…….